# Pericytes promote skin regeneration by inducing epidermal cell polarity and planar cell divisions

Lizhe Zhuang[1], Kynan T Lawlor[1], Holger Schlueter[1], Zalitha Pieterse[2], Yu Yu[2], Pritinder Kaur[1,2]

**The cellular and molecular microenvironment of epithelial stem/progenitor cells is critical for their long-term self-renewal. We demonstrate that mesenchymal stem cell–like dermal microvascular pericytes are a critical element of the skin's microenvironment influencing human skin regeneration using organotypic models. Specifically, pericytes were capable of promoting homeostatic skin tissue renewal by conferring more planar cell divisions generating two basal cells within the proliferative compartment of the human epidermis, while ensuring complete maturation of the tissue both spatially and temporally. Moreover, we provide evidence supporting the notion that BMP-2, a secreted protein preferentially expressed by pericytes in human skin, confers cell polarity and planar divisions on epidermal cells in organotypic cultures. Our data suggest that human skin regeneration is regulated by highly conserved mechanisms at play in other rapidly renewing tissues such as the bone marrow and in lower organisms such as *Drosophila*. Our work also provides the means to significantly improve *ex vivo* skin tissue regeneration for autologous transplantation.**

## Introduction

The self-renewal of many tissues occurs in the context of a cellular and molecular microenvironment better known as the niche, as originally postulated for the bone marrow (Schofield, 1978). In reality, tissue niches are complex with many interacting factors, including extracellular matrix proteins, tissue stiffness, growth factors, and their availability, regulating cell replacement and tissue architecture in concert with a variety of cell types, reviewed in depth recently (Xin et al, 2016). Although it is difficult to address all niche components at once, identifying the role of common elements found in tissues from different organs is likely to yield insights into conserved regulatory mechanisms that govern cell and tissue replacement.

The rapidly renewing epidermis of the human skin undergoes cell replacement in intimate association with its immediate dermal mesenchymal microenvironment. Indeed, its dependency on mesenchymal factors was evident from studies demonstrating that a feeder layer of embryonic fibroblasts was essential for epidermal cell/keratinocyte propagation in culture (Rheinwald & Green, 1975). Subsequent organotypic culture (OC) techniques for skin regeneration (Bell et al, 1981; Asselineau et al, 1986) confirmed that fibroblasts were critical for the more ordered spatial and temporal gene expression pattern observed in these three-dimensional skin equivalents, displaying keratinocyte proliferation in the basal layer and differentiation in the suprabasal layers (el-Ghalbzouri et al, 2002; Boehnke et al, 2007). However, the dermis of the skin is a complex and heterogeneous tissue with diverse functions, comprising several cell types, including dendritic, neural, endothelial, and immune cells and pericytes, in addition to fibroblasts. An understanding of the function of specific cell types and the molecular regulators that comprise the epidermal niche is essential to harnessing its regenerative potential for cell therapies. Attempts to dissect out those cells that support epithelial regeneration resulted in the identification of specialized dermal fibroblast subsets, that is, papillary and reticular dermal fibroblasts, defined by their proximity to the overlying epidermis. Papillary fibroblasts lie closer to the epidermis and appear to promote epidermal regeneration better than those from the deeper reticular dermis (Sorrell et al, 2004). In hair-bearing skin, dermal papilla fibroblasts found in the hair follicle base or bulb region and dermal sheath fibroblasts wrapped around the hair follicle with hair inductive capacity also support human interfollicular epidermal regeneration in both monolayer cultures (Hill et al, 2013) and OCs (Higgins et al, 2017).

Mesenchymal stem cell (MSC)–like populations derived from heterotypic tissues, specifically adipose-derived MSCs (Huh et al, 2007), also support epithelial regeneration in OCs. Our laboratory's attempts to identify cells found in the epidermal niche *in vivo* that influence human skin tissue renewal led to the discovery that dermal pericytes associated with microvessels close to the interfollicular epidermis, had the ability to improve epidermal

[1]Peter MacCallum Cancer Centre, Melbourne, Australia  [2]School of Pharmacy and Biomedical Sciences, Curtin Health Innovation Research Institute, Curtin University, Perth, Australia

Correspondence: pritinder.kaur@curtin.edu.au
Lizhe Zhuang's present address is University of Cambridge, MRC Cancer Unit, Cambridge, UK.
Kynan Lawlor's present address is Murdoch Children's Research Institute, Melbourne, Australia.
Holger Schlueter's present address is AstraZeneca, Lung Regeneration Bioscience, Gothenburg, Sweden.

regeneration in OCs (Paquet-Fifield et al, 2009), unrelated to their well-documented role in vascular structure and stability (Hirschi and DAmore, 1996; Armulik et al, 2005). We showed that dermal pericytes were potent MSC-like cells capable of conferring improved skin regenerative capacity on interfollicular keratinocytes that were already committed to differentiate, when combined with dermal fibroblasts, compared with fibroblasts alone (Li et al, 2004). Moreover, dermal pericytes not only expressed MSC markers but also had osteogenic, chondrogenic, and adipogenic differentiation capacity (Paquet-Fifield et al, 2009) in common with similar MSC-like cells that reside in the perivascular vessel wall in numerous organs (Crisan et al, 2008; Corselli et al, 2013). The observation that dermal pericytes promote epidermal regeneration is also consistent with the concept that bone marrow MSC-like pericytes are a critical element of haemopoietic stem cell niches supporting haemopoiesis both *in vitro* and *in vivo* (Morrison & Scadden, 2014; Birbrair & Frenette, 2016).

In this study, we further examined whether pericytes were sufficient for epidermal regeneration as the sole mesenchymal element and evaluated the quality of the resultant epithelial sheets. Our data demonstrate that pericytes were significantly better at maintaining a self-renewing epidermis conferring greater planar divisions within the proliferative compartment and a normal epidermal–dermal junction complete with hemi-desmosome and basement membrane assembly akin to normal skin, in contrast to dermal fibroblasts. Moreover, we provide evidence implicating BMP-2, a morphogenetic factor preferentially expressed by dermal pericytes (Paquet-Fifield et al, 2009), as a paracrine regulator of planar basal keratinocyte cell divisions.

# Results

### Dermal pericytes support the most normal epidermal tissue regeneration in OCs

We have previously shown that CD45$^-$VLA-1$^{bri}$ dermal cells are pericytes on the basis of their spatial location around dermal microvessels, the expression of a number of pericyte mRNAs (PDGFBR, NG-2/CSPG-4, αSMA, and RGS5) and requirements for specialized growth media (Paquet-Fifield et al, 2009). In contrast, CD45$^-$VLA-1$^{dim}$ dermal cells phenotypically resemble fibroblastic cells, expressing typical fibroblast mRNAs (platelet derived growth factor A, fibroblast activation protein α [FAPα], and fibroblast growth factors) and require significantly less stringent culture conditions (Paquet-Fifield et al, 2009). The distinct identity of these two mesenchymal cell types was further confirmed by determining the expression of FAPα (fibroblast activation protein α), a cell surface proteolytic enzyme identified in cultured fibroblasts (Scanlan et al, 1994) and expressed in fetal and newborn skin fibroblasts (Rettig et al, 1993). Immunostaining of skin tissue sections demonstrated that most of the mesenchymal dermal cells were FAPα positive, whereas VLA-1$^{bri}$ pericytes identifiable as αSMA positive were FAPα negative (Fig S1A and B). Moreover, flow cytometric analysis confirmed that the CD45$^-$VLA-1$^{dim}$ dermal fraction exclusively expressed FAPα (Fig S1C).

To investigate the effects of these two distinct cell types on epithelial tissue regeneration, we set up OCs as illustrated in

Fig S2A, plating freshly isolated human neonatal keratinocytes onto "dermal equivalents" (DEs) made up exclusively of CD45$^-$VLA-1$^{dim}$ fibroblasts, CD45$^-$VLA-1$^{bri}$ pericytes, or pericytes and fibroblasts at a ratio of 1:4, an incidence similar to that observed *in vivo*. Histological analysis of the resultant epithelial tissue demonstrated for the first time that dermal pericytes were indeed capable of providing the necessary signals for keratinocytes to undergo controlled proliferation and differentiation as the sole mesenchymal element of the dermis (Fig 1A, representative of three independent OC experiments). The epithelial tissue formed a spatially and temporally well-organized and stratified epidermis. Although the epithelial tissue obtained by using a solely fibroblastic (F) or combined fibroblast and pericyte (F + P) microenvironment also exhibited tissue regeneration, the epithelial sheets from F + P OCs were hyperproliferative with a thickened epidermis at a macroscopic level (Fig 1A), validated by quantitative analysis of three independent experiments (Fig 1B). Examination of the basal layer of the epithelial sheets revealed a highly polarized organization of cells within the basal layer solely in pericyte cocultures, with the closest similarity to native skin, unlike the F or F + P cocultures (Figs 1A and S2B; n = 3). Immunostaining for the proliferation marker Ki67 (Fig 1C) showed increased numbers of proliferative basal cells in the presence of pericytes or F + P compared with fibroblasts, confirmed quantitatively (Fig 1D). Immunostaining for ΔN-p63, a transcription factor with a dual role in maintaining epidermal cell proliferation and stratification associated with differentiation (Mills et al, 1999; Yang et al, 1999), demonstrated stronger and more extensive ΔN-p63 expression in pericyte cocultures basally and suprabasally (Fig 1E). Notably, the %Δ N-p63$^+$ basal cells were significantly higher in the OCs generated with pericytes alone (Fig 1F; n = 3 experiments). Moreover, K15, a marker of the skin's epidermal basal layer usually lost in activated hyperproliferative epithelia including psoriasis and OCs (Waseem et al, 1999) but re-expressed when they reach a presumably homeostatic state (Li et al, 2004), was expressed throughout the basal layer equivalent in pericyte-cocultured OCs—albeit at a low level—whereas its expression was sporadic in F + P or fibroblast OCs (Fig 1G), and virtually absent from fibroblast cocultured OCs.

Despite increased basal cell proliferation, the pericyte-cocultured keratinocytes displayed the most ordered spatial expression of the differentiation-specific keratin K10 (Fig 1H) compared with native skin, that is, all suprabasal layers were uniformly K10 positive, in contrast to its less uniform expression in epithelial sheets derived from cocultures with F or F + P. These data demonstrate that pericytes are not only sufficient for epidermal tissue regeneration but also secrete factors that promote the most normal basal and suprabasal layers in 3D cocultures.

### Dermal pericytes promote the most normal dermo-epidermal junction in OCs

Given the polarized appearance of the basal cells in OCs when cocultured with pericytes, combined with our previous observations that purified LAMA5 (found in the LN511/521 isoforms) was capable of promoting epidermal regeneration in OCs (Li et al, 2004) and that pericytes secrete LAMA5 in skin (Paquet-Fifield et al, 2009), we investigated the deposition of this extracellular matrix protein in OCs generated with pericytes, fibroblasts, or both by

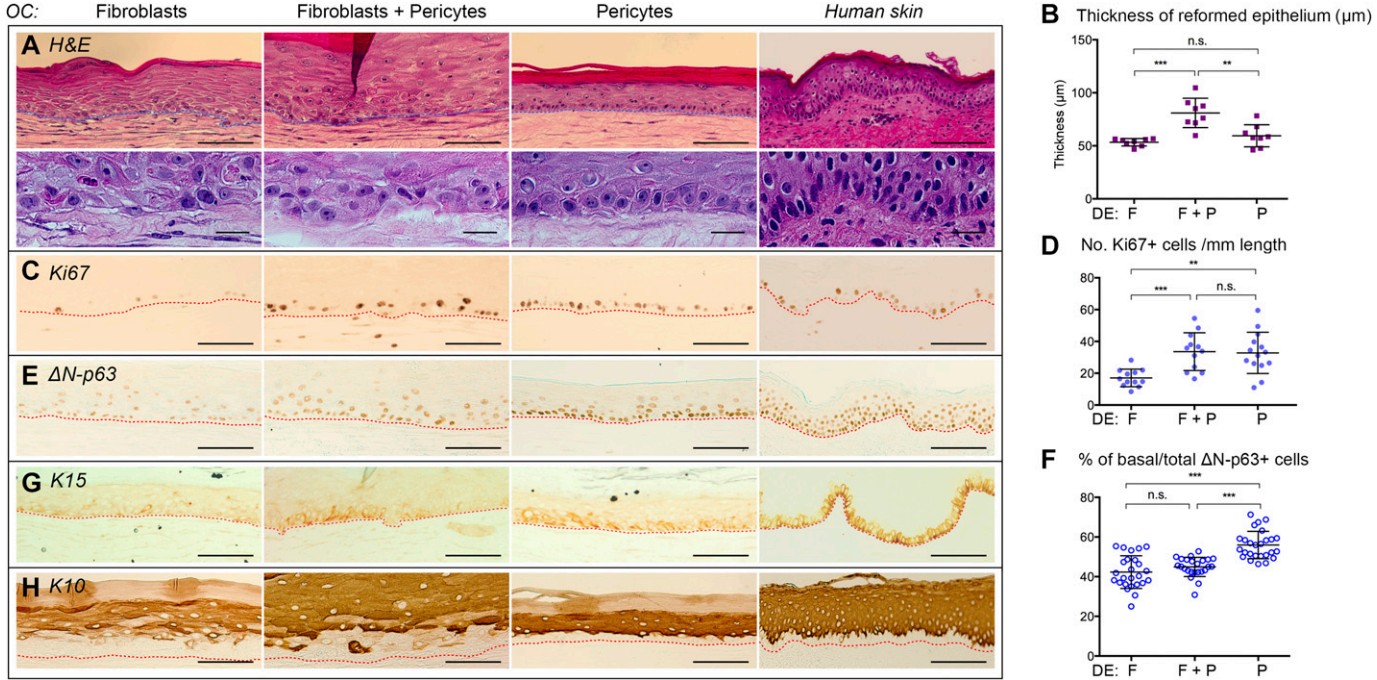

**Figure 1. Pericytes promote epidermal regeneration in OCs that most closely resembles human skin epidermis.**
**(A, C, E, G, H)** Human keratinocytes placed in OCs with DEs made up of fibroblasts, pericytes, or a combination of both exhibited differences in morphology. Hematoxylin and eosin staining revealed a polarized basal layer (A) and immunostaining showed increased expression of the proliferative markers Ki67 (C), ΔN-p63 (E), basal layer K15 expression (G), and ordered suprabasal K10 expression (H) in the presence of pericytes. Images representative of four independent OC experiments. **(B, D, F)** Quantitation of increased thickness of the epithelium (B), increased number of Ki67$^+$ cells (12 random fields/group) (D), and % basal/total ΔN-p63$^+$ (24 random fields/group) (F) in the presence of pericytes from four independent OC experiments. Error bars are mean ± SD. Unpaired $t$ test. **$P < 0.01, ***$P < 0.001, n.s. not significant. Scale bar = 100 $\mu$m.

immunostaining. Consistent with our previous published work, the use of fibroblasts alone resulted in sporadic LAMA5 deposition at the dermo-epidermal junction which was significantly improved by the inclusion of pericytes (Fig 2A F + P, representative of three independent OC experiments). However, uniform deposition of LAMA5 was closest to that observed in native skin in the pericyte-cocultured OCs (Fig 2A). Consistent with this data, ultrastructural analysis of replicate OCs revealed consistent basement membrane and hemi-desmosome assembly only in the presence of pericytes in every section examined (Figs 2B and S2C). In contrast, sporadic basement membrane and incomplete hemi-desmosome assembly were occasionally observed in the F + P OCs and rarely in fibroblast OCs (Figs 2B and S2C, data from two further replicate OC experiments). Confocal microscopy of whole mount OCs stained with phalloidin–rhodamine (actin fibers) and DAPI (nuclei) established that fibroblast-populated OCs had an irregular interface with the basal layer (Fig 3A) and that the fibroblasts retained a bipolar morphology (Fig 3B) evident from single scans in the dermo-epidermal junctional region. Moreover, z-stacks of sequential scans at 5-$\mu$m intervals taken from the dermal surface confirmed the more disorganized and irregular interface of fibroblast-populated OCs (Fig 3C). In contrast, pericyte-cocultured OCs showed close association with the basal layer (Fig 3D), consistent with the multiple processes extending from the stellate pericytes (Fig 3E). Unlike fibroblast-populated OCs, z-stack analysis of pericyte-populated OCs validated the presence of a more uniform, densely packed basal layer with an even interface between the epidermal and dermal compartments, and the proximity of pericytes to basal keratinocytes (Fig 3F).

## Pericytes promote increased planar keratinocyte divisions in the basal layer of OCs

Given that pericytes induced increased cell divisions and cell polarity within the basal layer, we set out to determine if the orientation of cell divisions in the basal layer of the OCs exhibited any differences when cocultured with pericytes versus fibroblasts. Thin (4-$\mu$m) histological sections of OCs were immunostained with DAPI to visualize DNA (Fig 3G), for phospho-histone H3 (pH3) to identify mitotic cells unequivocally (Fig 3H) and $\gamma$-tubulin to identify centrosomes which form part of the machinery that pulls chromosomes apart during mitosis (Fig 3I). The spindle pole or cell division orientation was determined by calculating the angle between the chromosomal axis (a virtual line joining the two centrosomes) and the dermo-epidermal junction. Analysis of the percentage of mitoses against the angle of division orientation (0° to 90°) from four independent OC experiments revealed that coculture with pericytes resulted in 45% of mitoses dividing in a plane parallel to the dermo-epidermal junction (<10° angle) presumably resulting in two daughter cells retained in the basal layer, compared with ~10% in fibroblast cocultures (Fig 3J). These data demonstrate that the cellular microenvironment of epidermal cells can influence the orientation of cell divisions—specifically pericytes are capable of increasing divisions in a plane parallel to the

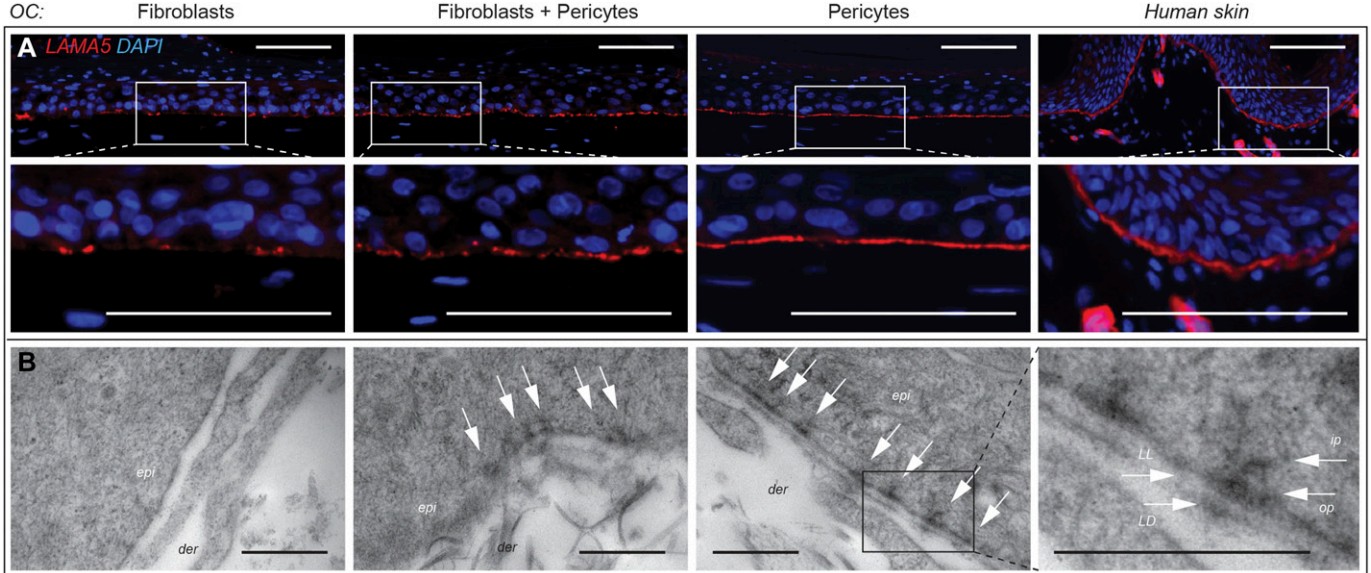

**Figure 2. Pericytes enhance LAMA5 deposition and assembly of the basement membrane and hemi-desmosomes in OCs.**
**(A)** Immunofluorescent staining for LAMA5 (red) and DAPI (blue, nuclei) in OCs populated with fibroblasts, pericytes, or a combination of both demonstrating its uniform deposition in pericyte-populated OCs. Images representative of three independent OC experiments. Scale bar = 100 $\mu m$. **(B)** Transmission electron micrographs of the dermo-epidermal junction of OCs revealing absence of a basement membrane in fibroblast OCs, sporadic initiation of hemi-desmosomes in fibroblast and pericyte-combined OCs, and complete basement membrane assembly with a continuous lamina lucida (LL) and lamina densa (LD) (enlarged boxed area) and regular hemi-desmosomes (white arrows) in pericyte OCs. The higher magnification pericyte OC transmission electron micrograph illustrates the inner plaque (ip) and outer plaque (op) of the hemi-desmosomes. Data representative of 18 random fields from two independent OCs. epi, epidermis; der, dermis. Scale bar = 0.5 $\mu m$.

basement membrane resulting in the production of more basal proliferative cells.

### Transduction of fibroblasts with BMP-2 increases epidermal cell polarity and planar cell divisions

We next set out to determine whether pericyte-secreted factors may regulate the plane of keratinocyte cell divisions in OCs by selecting a number of mRNAs preferentially expressed by pericytes but not fibroblasts from microarray analysis (Paquet-Fifield et al, 2009) that encoded secreted proteins, including GPX3, BMP-2, CCL8, ANGPT2, A2M, EGFL6, and platelet derived growth factor A. cDNAs corresponding to these genes were cloned into an internal ribosome entry site-GFP lentiviral vector and the virus generated was used to transduce human foreskin fibroblasts. GFP expression in the transduced fibroblasts was confirmed by fluorescence microscopy (Fig S3A) and BMP-2 transcripts validated by quantitative PCR (q-PCR) (Fig S3B). Demonstrating BMP-2 secretion from pericytes and BMP-2–transduced fibroblasts in either conditioned media or cell lysates by enzyme linked immunosorbent assay proved difficult, despite a well-validated enzyme linked immunosorbent assay detecting recombinant BMP-2 in the picogram range and Western blots detecting recombinant BMP-2 (data not shown). We attribute this to the demonstrably low levels of BMP-2 transcripts even in pericytes demonstrated by microarray analysis (Paquet-Fifield et al, 2009). Although in situ immunofluorescence detection of BMP-2 has been reported in the literature in many cell lines, to convincingly demonstrate its expression in primary cultured pericytes, pretreatment with brefeldin A, which inhibits protein secretion via accumulation of Golgi-resident proteins in the

endoplasmic reticulum (Klausner et al, 1992), was required to visualize it by in situ immunofluorescence (Fig S3C and D), otherwise undetectable in untreated cells (Fig S3E and F). Brefeldin A treatment of control GFP-transduced fibroblasts did not result in BMP-2 detection (F_GFP, Fig S3G and H; representative of three independent experiments), whereas similar treatment of BMP-2–transduced fibroblasts (F_BMP2, Fig S2I and J) allowed us to detect extremely low levels of BMP-2 compared with pericytes. Morphometric quantitation of the BMP-2 immunofluorescence signal per cell using Image J software verified that F_BMP2 cells had statistically significantly lower levels of BMP-2 than pericytes ($P < 0.0001$) but higher levels than those detected in F_GFP control fibroblasts ($P < 0.01$) (Fig S3K).

Subsequently, we incorporated F_BMP2 or control F_GFP–transduced fibroblasts into OCs to ascertain if they could recapitulate the pericyte effects on keratinocytes with respect to epidermal thickness, cell polarization, and plane of cell divisions. Notably, OCs populated with F_BMP2 fibroblasts gave rise to thicker epithelial sheets than OCs regenerated with F_GFP control fibroblasts (Fig 4A), verifiable by quantitation from three independent experiments (Fig 4B). Importantly, F_GFP control cells did not exhibit a polarized basal layer (Fig 4A, F_GFP, data representative of three independent experiments) as observed with untransduced fibroblasts (Fig 1A). In contrast, F_BMP2 cells restored polarity in the basal layer (Fig 4A, F_BMP2, representative of three independent experiments) and showed stronger staining for the proliferative marker Ki67 (Fig 4C), although the number of Ki67+ cells remained unchanged in F_GFP versus F_BMP2 OCs (Fig 4D). In comparison, ΔNp63 staining was not only stronger in the basal layer of F_BMP2 OCs compared with F_GFP control OCs (Fig 4E) but revealed statistically significantly higher

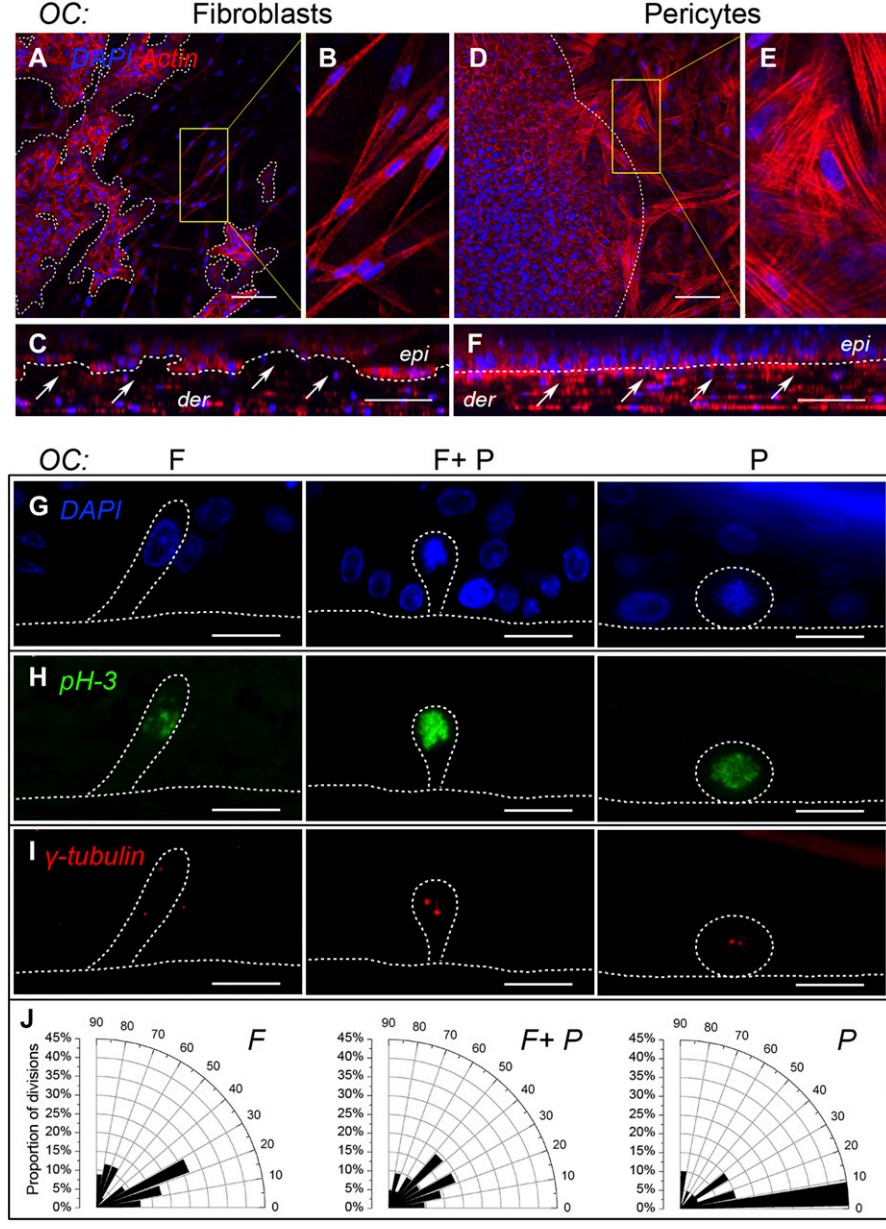

**Figure 3. Pericytes make close contact with the basal layer and instruct basal keratinocytes to undergo predominantly planar divisions parallel to the basement membrane in OCs.**
**(A–F)** Whole mount confocal microscopy of OCs populated with fibroblasts (A–C) or pericytes (D–F) stained with rhodamine-conjugated phalloidin (red, actin) and DAPI (blue, nuclear); z-stacks (C, F) reveal irregular dermo-epidermal junctions in fibroblast-populated OCs (A, C) and their bipolar morphology (B). Pericyte-populated OCs show stellate pericyte morphology (E) and their proximity at the dermo-epidermal interface (D); z-stack reveals a well-organized and even basal layer and close contact with pericytes (F). Images representative of six random fields from one of three OC experiments. White dotted line = interface between the epidermis (epi) and dermis (der). Scale bar = 100 $\mu$m. **(G–I)** Triple immunofluorescent staining for DAPI (G: blue, DNA), pH 3 (H: green, mitotic cells), and $\gamma$-tubulin (I: red, centrosomes) in OCs populated with fibroblasts (F), pericytes (P), or a combination of both (F + P), showing representative cell divisions. Cell division orientation relative to the basement membrane was determined by measuring the angle between the dermo-epidermal junction and the centrosomal axis in pH 3[+] cells. J. Quantitation of the angle of cell division (0°–90°) as % of total mitotic events in OCs populated with fibroblasts (F), pericytes (P), or both (F + P), from 43, 23, and 23 mitoses per group from three independent experiments displayed as a radial bar chart showing a tendency for mitoses parallel to the basement membrane in the presence of pericytes. Scale bar = 20 $\mu$m.

numbers of $\Delta$Np63[+] cells (Fig 4F, $P$ < 0.05, n = 3). Immunostaining revealed increased LAMA5 deposition at the dermo-epidermal junction (Fig 4G) and better spatial K10 expression (Fig 4H) in OCs with F_BMP2 fibroblasts compared with F_GFP controls—although this was not restored to the levels observed with pericyte inclusion in the OCs (Figs 1D and 2A). Interestingly, K15 expression remained absent within the basal layer of OCs regenerated by either BMP-2– or GFP-transduced fibroblasts (Fig S3L). However, despite these shortcomings, the introduction of BMP-2–transduced fibroblasts in OCs resulted in the restoration of greater numbers of planar cell divisions parallel to the dermo-epidermal junction, that is, >35% at an angle of <10° and >65% at an angle of <20° (Fig 4I–K), contributing to the greater numbers of proliferative epidermal basal cells compared with about 10 and 25% at an angle of <10° and <20°, respectively, to

the basement membrane in vector GFP–transduced control fibroblasts (Fig 4L–N), similar to untransduced fibroblasts (Fig 3I, ~10% at <10° and ~17.5% at <20°).

## Discussion

Early studies revealed that pericytes had a major role in regulating vascular functions, including vessel dilation, vascular permeability, and integrity (Gerhardt & Betsholtz, 2003). Recent studies suggest that pericytes exhibit stem cell properties with multi-lineage mesenchymal differentiation potential (Crisan et al, 2008) contributing to bone, cartilage, and muscle tissue regeneration in

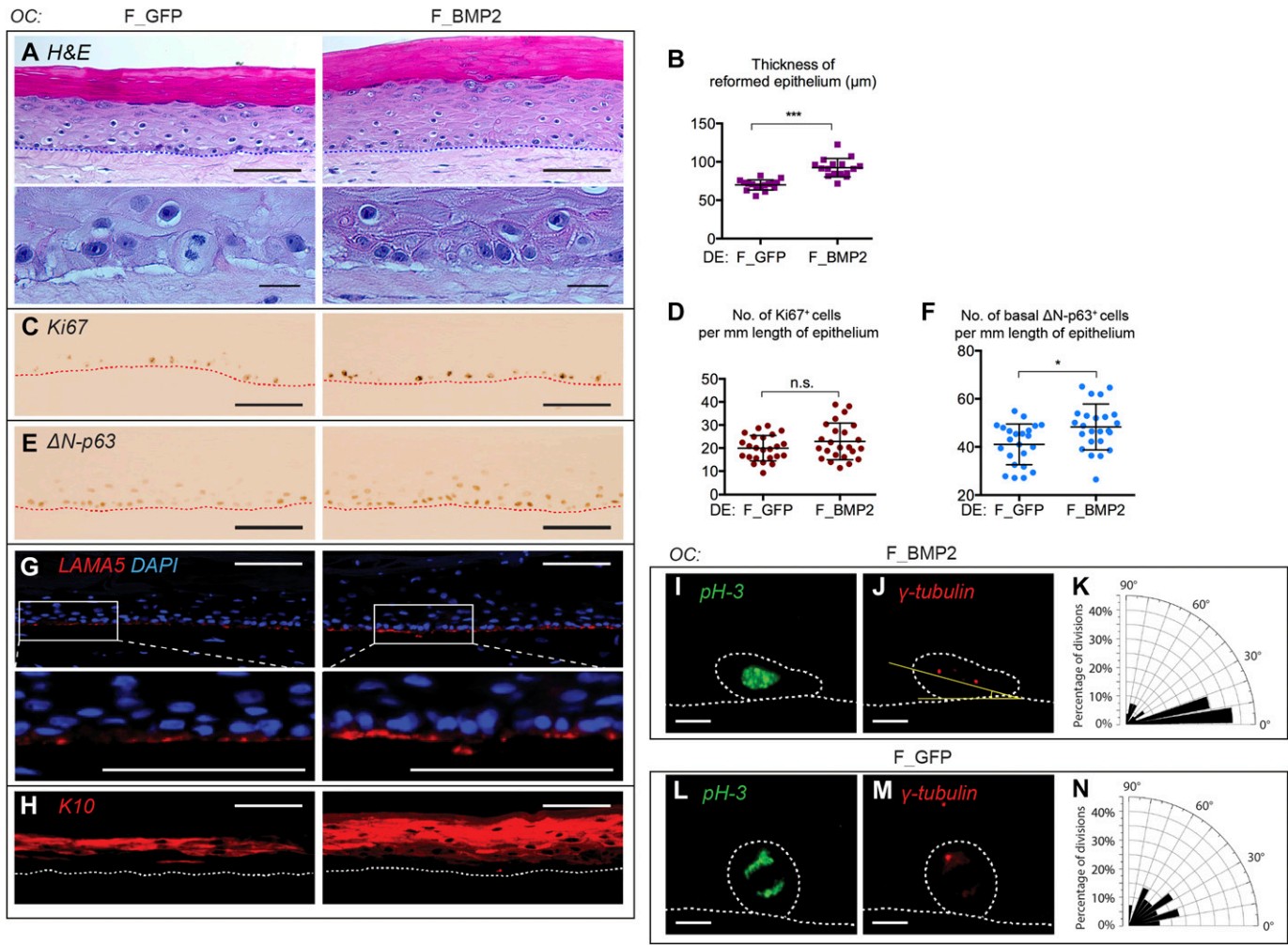

**Figure 4.   Fibroblasts expressing BMP-2 increase epithelial thickness, restore cell polarity, increase planar cell divisions, and ordered differentiation to human keratinocytes in OCs.**
**(A, C, E, G, H)** OCs populated with control (F_GFP) or BMP-2–transduced (F_BMP2) fibroblasts were analyzed after hematoxylin and eosin staining (A) or immunostaining with Ki67 (C), ΔN-p63 (E), LAMA5 (G), and K10 (H) to reveal improved basal cell polarity, increased expression of Ki67, ΔN-p63, and LAMA5 and better suprabasal K10 expression in the presence of F_BMP2 compared with F_GFP controls. Images are representative of three independent experiments. **(B, D, F)** Quantitation of epithelial thickness (B), number of Ki67+ (D), and ΔN-p63+ (F) cells in OCs populated with control (F_GFP) or BMP-2–transduced fibroblasts (F_BMP2). Error bars represent mean ± SD; statistical analysis performed using unpaired $t$ test; n.s. not significant; $*P < 0.05$; $***P < 0.00$. All quantitative data are derived from three independent experiments. **(I, J, L, M)** Immunostaining of OCs for pH 3 (green) and γ-tubulin (red) to determine keratinocyte cell division orientation in the presence of BMP-2–transduced (I, J: F_GFP, n = 28 mitoses) and control fibroblasts (L, M: F_BMP2, n = 27 mitoses). **(K, N)** Radial bar charts illustrating angle of keratinocyte mitoses obtained with F_BMP2 (K) and F_GFP control (N) demonstrating increased planar divisions in the latter.

experimental settings (Sa da Bandeira et al, 2017). Our previous (Paquet-Fifield et al, 2009) and current work point to a novel, paracrine role for pericytes in influencing the quantitative and qualitative cell and tissue regenerative capacity of human keratinocytes. Parallel observations have been made in the bone marrow, demonstrating that pericytes are an integral part of the haemopoietic stem cell niche regulating their maintenance and quiescence through paracrine effects (Sacchetti et al, 2007), reviewed in Sa da Bandeira et al (2017).

The dependency of human keratinocytes on mesenchymally derived paracrine regulators for both cell replicative and tissue regenerative functions has been evident for several decades (Rheinwald & Green, 1975; Asselineau et al, 1986; el-Ghalbzouri et al, 2002; Boehnke et al, 2007; Bell et al, 1981). However, our data provide

an insight into the potential role of a specific subset of MSC-like perivascular dermal cells, that is, pericytes on epidermal renewal in human skin. The possibility that dermal pericytes could augment the paracrine effect of fibroblasts on epithelial regeneration was first revealed by us previously, unexpectedly demonstrating a pro-epidermal regenerative function in 3D OCs, unrelated to their well-documented role in the vasculature (Paquet-Fifield et al, 2009). In that study, pericytes were co-inoculated with fibroblasts in the "dermal equivalent" or microenvironment of keratinocytes with intrinsically poor proliferative and tissue-regenerative potential (i.e., non-stem cells) resulting in vastly enhanced tissue reconstitution. Given the existence of perivascular cells in the haemopoietic system that appear to be vital to haemopoietic tissue and stem cell self-renewal (Sa da Bandeira et al, 2017), we set out to

determine whether dermal perivascular cells—pericytes—had a similar function in the skin, in comparison with the more widely used dermal fibroblasts. Unexpectedly, we found that dermal pericytes appear to influence skin regeneration to a more homeostatic level exhibiting both morphological and molecular properties closer to normal skin than when cocultured with fibroblasts. Hence, the tissue architecture of the epidermal sheets obtained in OCs containing pericytes displayed more organized cell polarity within the proliferative compartment and more ordered suprabasal stratification, concomitant with higher expression levels of ΔN-p63, a transcription factor implicated not only in epidermal stratification (Koster et al, 2004) but also in maintaining a higher proliferative potential in basal keratinocytes (Yang et al, 1999). Notably, the basal layers of epidermal sheets generated by pericyte coculture not only had the highest number of ΔN-p63[+] cells but also the greatest proliferative index identifiable as Ki67[+] cells. Importantly, p63 is also implicated in promoting asymmetric cell divisions given that p63 null mice exhibit decreased perpendicular epidermal cell divisions (Lechler & Fuchs, 2005). Our data suggest that pericyte coculture promotes p63 expression and, thus, increased asymmetric divisions and proper stratification. Moreover, the re-establishment of epidermal homeostasis by pericytes was evident by the expression of K15 in the basal layer of pericyte-populated OCs, whereas this was not the case for fibroblast or F + P OCs. Concomitantly, despite an increase in proliferative index, the differentiation program of the pericyte-cocultured keratinocytes was unperturbed, and indeed spatial expression of K10 suprabasally was normalized compared with its more disorganized expression with fibroblasts. Notably, the effect of combining pericytes and fibroblasts was interesting—although hyperproliferation was evident histologically (Fig 1A and B) consistent with an increase in the Ki67[+] cells (Fig 1D), there was no change in the %ΔN-p63[+] basal keratinocytes (Fig 1F) consistent with the low number of planar divisions (Fig 3J) and disorganized differentiation and stratification (Fig 1H). Interestingly, the overall proliferative and differentiation changes observed with a combination of fibroblasts and pericytes in the DE as compared with fibroblasts or pericytes alone indicate that fibroblasts appear to inhibit the effects of pericytes. This may be mediated by alterations in the profile of secreted growth factors, meriting future investigation. These data highlight the distinction between increased overall proliferation attainable by pro-proliferative signals versus qualitative changes to increased planar divisions leading to homeostatic cell replacement in the proliferative compartment without interference to ordered differentiation.

The most exciting observation we report was the ability of extrinsic dermal cues provided by pericytes to regulate the orientation of overlying keratinocyte cell divisions in the proliferative basal cell compartment of the epidermis. In the epidermis, planar, presumed symmetric divisions occur parallel to the basement membrane, generating two basal cells, whereas asymmetric divisions occur perpendicular to it, resulting in their placement in distinct niches within the tissue: one adjacent to the basement membrane in the basal layer and the other suprabasally, resulting in signals that promote cell proliferation versus differentiation, respectively. Thus, the balance of symmetric versus asymmetric divisions within the basal layer is critical for the maintenance of homeostatic balance within the interfollicular epidermis. Much of what we know about asymmetric cell divisions in the interfollicular epidermis comes from studies in genetic mouse models which link the cellular machinery, particularly polarity proteins with orientation of mitotic spindles and epidermal stratification during embryogenesis. Thus, in epidermal development, mitotic spindle orientation and thus cell division orientation changes from predominantly symmetric to largely asymmetric, to generate a stratified epithelium (Lechler & Fuchs, 2005). Perturbation of highly conserved cell polarity proteins, including inscuteable/mInsc, Pins/leu-gly-asn, and aPKC that drive mitotic spindle orientation, as well as NuMA and dynactin can result in stratification defects due to incorrect spindle orientation (Williams et al, 2011, 2014; Tellkamp et al, 2014). A recent 3D analysis of cell division orientation in adult murine skin suggests that most of the cell divisions in thinner epithelia such as dorsal and ear epidermis occur parallel to the basement membrane, whereas thicker epithelia such as hind paw and tail epidermis which most closely resemble the thicker human epidermis contain both parallel and oblique orientations (Ipponjima et al, 2016). Consistent with our findings, analysis of human epidermis reconstituted in OCs also revealed parallel and oblique cell division orientations (Noske et al, 2016).

It could reasonably be argued that lineage tracing or live cell imaging of dividing epidermal cells is required to be certain of the fate of daughter cells that are located suprabasally during mitoses but remains beyond the limits of technology at present. However, authoritative work in spindle orientations and epidermal cell fate indicates that the nature of perpendicular spindle orientations within the basal layer with one cell remaining in contact with the basement membrane and the other losing contact with it automatically result not only in stratification but also in asymmetric cell divisions (Lechler & Fuchs, 2005). These investigators argue that because many key functions (i.e., extracellular matrix protein secretion, integrin-mediated focal adhesion, and growth factor signaling) are known to be basally localized, perpendicular divisions provide a natural mechanism for their unequal partitioning to two daughter cells. In addition, elegant live tissue imaging studies of murine skin from the Greco Lab at Yale show that once a dividing cell migrates suprabasally, migration back into the basal layer is not observed (Rompolas et al, 2016). It is important to note that the individual fate of daughter cells resulting from mitosis parallel to the basement membrane may vary—indeed, the Greco Lab have shown that daughter cells can have independent fates and neither is more or less likely to stay in the basal layer or differentiate (Rompolas et al, 2016). These data combined with our observations correlating increased cell divisions parallel to the basement membrane with increased Ki67, ΔNp63, LAMA5, and K15 expression together with the deposition and assembly of a basement membrane and hemidesmosomes which undoubtedly indicate normal basal cell polarity in the presence of pericytes strongly support the concept that these cells indeed confer symmetric divisions on basal keratinocytes.

As shown by us previously, the deposition of LAMA5 at the dermo-epidermal junction was improved in pericyte-reconstituted OCs; moreover, the assembly of a uniform ultrastructurally demonstrable basement membrane and adhesive hemi-desmosomes occurred only in the presence of pericytes pointing to their ability to restore homeostasis to a greater extent than fibroblasts. In this context, it is noteworthy that LN511, an LAMA5 isoform of laminin

maintains human embryonic stem cells in a more primitive un-differentiated state in the absence of feeder layers in culture (Rodin et al, 2010). The close physical proximity of the pericytes to the basal keratinocytes in the OCs demonstrated by whole mount analysis (Fig 3A–F) suggests that pericytes directly contribute to basement membrane matrix components, specifically LAMA5, consistent with our previous immunogold localization data demonstrating that dermal pericytes in human skin synthesize and secrete LAMA5 *in vivo* (Paquet-Fifield et al, 2009). Indeed, basal keratinocyte polarization in pericyte-populated OCs can be attributed at least in part to the deposition of LAMA5 and basement membrane assembly which set up apical basal polarity in the basal layer. Basement membrane assembly may be critical in regulating asymmetric cell divisions, given that $\beta$1 integrin–null mice, incapable of basement membrane assembly, display inappropriate localization of the leu-gly-asn-mlnsc apical polarity complex and abnormalities in spindle orientation resulting in random cell division orientations in the epidermis (Lechler & Fuchs, 2005).

Additional extrinsic cues have been described that implicate Wnt/BMP/FGF signaling in driving epidermal stratification. Para-crine Wnt signaling is required for the maintenance of proliferative basal cells and epidermal stratification in embryonic skin as evidenced by epidermal specific knockout of Gpr177, a G protein–coupled Wnt ligand receptor (Zhu et al, 2014). Gpr177 mediates all intracellular Wnt signaling and its absence results in a severely compromised proliferative basal layer in embryonic limb skin. Interestingly, that study also demonstrated that epidermal Gpr177 deletion led to decreased expression of bone morphogenetic protein (BMP) 2, 4, and 7 in the developing epithelium and mes-enchyme—and further that disruption of Wnt-dependent down-stream BMP signaling in the dermis was responsible for the proliferative defects in the epidermis of Gpr177 KO mice (Zhu et al, 2014). It was further demonstrated that BMP signaling via pSmad1/5/8 led to transcriptional activation of FGF-7 and FGF-10 in dermal cells which acted on keratinocytes via FGFRII. These data are in-teresting in light of our observation that dermal expression of BMP-2 demonstrated by us previously to be preferentially expressed by pericytes was capable of restoring planar divisions within the basal layer when introduced into fibroblasts, otherwise incapable of promoting these types of cell divisions. Consistent with the low BMP-2 transcript levels observed by us previously in pericytes (Paquet-Fifield et al, 2009), BMP-2 protein levels were also de-monstrably low in these cells (Fig S3C). Given that BMP-2 protein levels were even lower in BMP-2–transduced fibroblasts (Fig S3I), it remains possible that increased levels of this morphogen could elicit a greater increase in planar or parallel cell divisions than we observed. Similarly, it remains unclear whether the partial resto-ration of LAMA5 expression and the absence of K15 can be attributed to insufficient levels of BMP-2 expression in the transduced fibroblasts—further work with varying levels of BMP-2 expression is required. Furthermore, unequivocal evidence in support of a role for BMP-2 in influencing spindle pole or cell division ori-entation in keratinocytes, such as BMP-2 knockdown in pericytes, is essential.

There is precedence for BMPs being important in regulating asymmetric cell divisions in *Drosophila* related to gonadal stem cell fate and maintenance. Positional placement of gonadal stem cells adjacent to somatic hub cells ensures stem cell maintenance mediated by *dpp* and *gbb*—two members of the *Drosophila* BMP family expressed by the somatic cells (Kawase et al, 2004). These studies together with our results argue for a highly conserved mechanism of retaining stem and progenitor cells within a tissue in both invertebrates and vertebrates. We therefore speculate that in the interfollicular epidermis, pericytes arranged in a nonrandom pattern within the rete ridges of the dermis provide regulatory signals, including BMP-2, that result in predominantly symmetric divisions in the overlying epidermis, whereas regions enriched in fibroblasts confer predominantly asymmetric divisions driving basal cells into the suprabasal layer leading to differentiation. Notably, a recent study in murine skin showed that proliferative epidermal stem cells were located close to blood vessels, and therefore, by default near pericytes (Sada et al, 2016). We hy-pothesize that the net regional stromal content of the dermis (pericyte:fibroblast ratio) determines the balance of symmetric versus asymmetric divisions in the epidermis and that pericyte-enriched regions promote maintenance of basal keratinocytes within the stem and progenitor compartment consistent with their ability to retain cells attached to the basement membrane. In-terestingly, a study in *Drosophila* ovary showed that Bmp signaling from niche cells repressed differentiation of gonadal stem cells (Song et al, 2004). Our hypothesis suggests heterogeneity within the basal layer with respect to the placement of stem and progenitor cells versus committed cells destined to migrate suprabasally and with respect to their location within the rete ridge structures found in human skin—a notion with some basis in the literature. In murine epithelia, it appears that all basal cells are equipotent supported not only by *in vivo* lineage tracing studies (Clayton et al, 2007) but also elegant work combining lineage analysis with fate mapping by intravital cell imaging in whole animals (Rompolas et al, 2016). However, murine and human skin have distinct tissue architecture related to form and function that most likely influences tissue renewal. Indeed, a recent study in a human patient transplanted with genetically modified and, therefore, trackable epidermal cells indicates that not all clonogenic cells are capable of sustaining skin renewal over time, arguing against the equipotency of basal cells in human skin (Hirsch et al, 2017).

In conclusion, our study suggests that pericytes may be more suitable as feeder cells for the *ex vivo* expansion of keratinocytes before autologous transplantation, while simultaneously deepening current understanding of the inductive mesenchymal microenviron-ment of the human epidermis. It also constitutes the first report describing an extrinsic cue driven by a previously unsuspected cell type present in the epidermal microenvironment that modulates mitotic spindle orientation, resulting in a more proliferative basal layer.

## Materials and Methods

### Isolation and culture of keratinocytes, dermal pericytes, and fibroblasts

Human neonatal (1- to 4-week old) foreskin tissue was collected from routine circumcision with informed consent from guardians. Foreskin tissue was processed to obtain epithelial keratinocytes

and dermal cells as described (Gangatirkar et al, 2007). 1 × 10⁷ cells/ml dermal cells were stained for VLA-1 (1:20, MCA1133F; Serotec) and CD45 (1:40; 555483; BD Biosciences) for an hour and counterstained with propidium iodide (1:200, P3566; Life Technologies) to permit exclusion of dead cells before sorting for VLA-1$^{bri}$CD45$^−$ dermal pericytes and VLA-1$^{dim}$CD45$^−$ fibroblasts as described (Paquet-Fifield et al, 2009) and fractions reanalyzed to verify purity. Pericytes and fibroblasts were cultured in EGM-2 (CC-3162; Lonza) and DMEM containing 10% fetal bovine serum (DMEM-10), respectively, at 37°C with 5% $CO_2$. All experimentation was approved by the Institutional Human Research Ethics Committee (Project: 03/44).

## OC

OC was conducted as described (Gangatirkar et al, 2007) with the following modifications: cotton pads (Organogenesis Inc.) were replaced by absorbent pads (AP1002500; Millipore), T3 (triiodothyronine) was prepared at 1 $\mu$M in the ITT aliquots to obtain 2 nM final concentration, and L-glutamine was replaced by GlutaMAX (35050061; Life Technologies). Bovine collagen type I was a kind gift from Organogenesis Inc.

Briefly, cultured pericytes or fibroblasts were harvested at ~90% confluence to prepare DEs containing a total of 7.5 × 10⁴ pericytes or fibroblasts per DE, or both fibroblasts and pericytes at a 4:1 ratio reflecting their incidence in neonatal human dermis. 5 × 10⁴ primary uncultured human neonatal keratinocytes resuspended in 30 $\mu$l of EpiLife (M-EPI-500-CA and S-001-5; Life Technologies) were seeded on top of each DE. OCs were maintained in epidermalization medium for a week and then at an air–liquid interface in cornification medium for a week and then in maintenance medium for another week before harvest.

### Immunostaining

OCs and skin tissue (~12 × 3 mm pieces) were fixed in 10% neutral buffered formalin (45 min at RT) and paraffin-embedded to obtain 4-$\mu$m sections. The sections were de-waxed and hydrated before performing antigen retrieval in a pressure cooker at 125°C for 3 min in Tris–EDTA buffer, pH 9.0. The sections were stained for Ki67 (1:200, M7240; Dako), ΔN-p63 (1:400, 619001; BioLegend), K15 (1:50, ab2414; Abcam), K10 (1:300, ab9029; Abcam), p-H3 (1:300, 641001; BioLegend), γ-tubulin (1:2,000, T5326; Sigma-Aldrich), and PDGFRβ (1:50, ab32570; Abcam). Immunohistochemistry or immunofluorescence was performed using ImmPRESS kits (Vector Laboratories) or 1 $\mu$g/ml DAPI and fluorophore-conjugated antibodies (A-21241, A-21134, A-21428, A-11006, or A-21124; Life Technologies) at 1:200 for 1 h at RT. For LAMA5 detection, paraformaldehyde-fixed tissue was embedded in optimal cutting temperature (Sakura) and 10-$\mu$m sections cut on a cryostat (Leica Biosystems) at a temperature setting of object temperature = −20°C; chamber temperature = −20°C. The sections were stained for LAMA5 with neat, hybridoma supernatant (clone 4C7) and secondary fluorescent antibody. To determine cell division orientation in OCs, the sections were simultaneously stained with phospho-histone-3 (antibody) and γ-tubulin (antibody) and DNA counterstained with DAPI.

For whole mount analysis, OCs were fixed in 4% paraformaldehyde (25 min at RT), cut into 3 × 6 mm pieces, permeabilized in phosphate buffered saline with Tween-20 (PBST) for 1 h and then incubated in 1 $\mu$g/ml DAPI and 1 U/ml rhodamine-conjugated phalloidin to stain nuclei and actin filaments. OCs were rinsed in PBST three times

before incubation in PBST (1 h) and 50% glycerol (1 h). The OC pieces were trimmed to 2 × 3 mm, mounted in 90% glycerol, and scanned on a Nikon C2 Confocal Microscope System from the dermal surface at 5-$\mu$m intervals.

### Transmission electron microscopy

OC pieces (3 × 6 mm) were fixed in 2.5% glutaraldehyde, 2% paraformaldehyde in 0.1 M cacodylate buffer for 2 h at RT and washed in cacodylate buffer (3 × 10 min), postfixed in 2% OsO4 and the washes repeated. Tissues were rinsed in distilled water (3 × 10 min), dehydrated in ethanol, treated with acetone (2 × 30 min), followed by 1:1 Acetone/Spurr's resin (2 × 2 h) before impregnation with 100% Spurr's resin (2 × 2 h) under vacuum, and embedded for ultrathin sectioning. Transmission electron microscopy was performed using a JEOL 1011 (JEOL USA, Inc.).

### Lentivirus generation, verification, and transduction

pLV411G plasmid was a gift from Dr. Simon Barry (University of Adelaide). pLV411G-carrying human BMP-2 and/or vector control cDNA were cloned (Skalamera et al, 2012) and verified by sequencing. 5 × 10⁵ HEK293T cells were plated and cultured in DMEM-10 without antibiotics until 60% confluent before lipofection with 5 $\mu$g DNA consisting of 2.6 $\mu$g of pLV411G, 0.6 $\mu$g of VSV-G, 1.26 $\mu$g of Gag/Pol, and 1.06 $\mu$g of Rev (Addgene) with Lipofectamine Reagent (Invitrogen) for 24 h at 37°C, 5% $CO_2$. The cells were then washed with PBS and maintained in 2.4 ml DMEM-30 for 2 d to allow viral particle secretion into the medium. A second batch of viral supernatant was collected by culturing the transfected cells in another 2.4 ml DMEM-30 for 2 d. Viral supernatant was filtered (0.2 $\mu$m filter; Millipore) and stored at −80°C.

Freshly sorted VLA-1$^{dim}$/CD45$^−$ neonatal fibroblasts were plated at ~5 × 10⁵ cells per well in DMEM-10 in six-well plates. The cells were transfected at ~80% confluence by incubating with 2 ml/well of the first batch of viral supernatant containing 10 $\mu$g/ml polybrene for 15–20 h; and with the second batch of viral supernatant and expanded in vitro. Stable GFP expression was monitored regularly by fluorescence microscopy and the GFP⁺ fraction enriched by fluorescence activated cell sorting prior to expansion before use in OCs. Lentiviral vector usage was approved (NLRD #09/2007) by the Gene Technology Regulator Office of the Australian Government.

The relative expression of BMP-2 in the transduced human fibroblasts was verified using real-time reverse transcriptase qPCR. mRNA was extracted from cultured transduced fibroblasts (p6) using an RNeasy Mini kit (Qiagen). 2 $\mu$g of mRNA was used in the Reverse Transcription System (Promega) to generate cDNA. qPCR was conducted in a StepOnePlus Real Time PCR System (Applied Biosystems) in 20 μl containing 10 ng cDNA, 125 nM forward/reverse primers, and 10 μl of Fast SYBR Green Master Mix using the program cycle: 95°C, 10 min; 95°C, 15 s; 60°C, 1 min, ×40; 95°C, 15 s; 60°C, 1 min; and 95°C, 15 s. The relative expression of BMP-2 was calculated using ΔΔCt and normalized against GAPDH. 5′–3′ primer sequences were as follows: *BMP-2 forward primer: TCCTGAGCGAGTTCGAGTTG, BMP-2 reverse primer: CCAAAGATTCTTCATGGTGGAAGC, GAPDH forward primer: GTGAAGGTCGGAGTCAACG, and GAPDH reverse primer: TGAGGTCAATGAAGGGGTC.*

### BMP-2 protein expression

Pericytes, F_BMP2 and F_GFP cells were cultured on coverslips in their respective media and grown to 80% confluency. The medium was removed and the cells washed with PBS before incubation in pericyte or fibroblast media containing 5 $\mu$g/ml brefeldin A (Cat no. 420601, lot no. B236598; BioLegend) for 16 h at 37°C, 5% $CO_2$. The medium was then removed, cells washed with PBS, and then fixed in 4% para-formaldehyde for 20 min, followed by permeabilization with 0.1% saponin in PBS for 15 min at RT in the dark. The cells were then blocked for 1 h in 10% donkey serum in 0.1% saponin buffer and incubated overnight at 4°C either with rabbit polyclonal anti-BMP-2 (Cat no. Ab82511, Lot no. GR128280-1; Abcam) or rabbit IgG isotype control polyclonal antibody (Cat no. 02-6102, Lot no. SE252928; Thermo Fisher Scientific) at 5 $\mu$g/ml in blocking buffer. The cells were then washed three times with 0.1% saponin buffer (10 min each) and incubated in 5 $\mu$g/ml secondary antibody (Donkey Anti-Rabbit Alexa Fluor 568, Cat no. A10042, Lot no. 1476640; Life Technologies ) in 0.1% saponin buffer at RT in the dark for 1 h. The cells were washed three times as before, counterstained with DAPI, rinsed, and mounted with ProLong Gold Antifade mounting media. Slides were imaged on a Nikon A1 confocal microscope with a 60× water immersion objective lens. Individual cells were analyzed for fluorescence intensity using Image J, and statistical analysis was performed on GraphPad Prism 6 using one-way ANOVA with Tukey's multiple comparison test.

### Quantification and statistics

The length ($\mu$m) of the selected regions was obtained using ImageJ. Quantitative results from immunostaining and OC experiments were replicated in at least three independent experiments, and representative samples were displayed when reproducible results were obtained. Statistical analysis was performed using unpaired $t$ test. $P < 0.05$ was considered statistically significant. Error bars represent SD. All statistical analyses were performed using the software Prism version 6.0.

## Supplementary Information

## Acknowledgements

We are grateful to Nathan Godde for his advice on the analysis of cell division orientation, Sarah Ellis for assistance with fluorescence and transmission electron microscopy, and Ralph Rossi and colleagues for fluorescence activated cell sorting. This work was funded by National Health and Medical Research Council of Australia Project grant no. 1043453 to P Kaur and an Australian Postgraduate Award PhD scholarship to L Zhuang.

### Author Contributions

L Zhuang: conceptualization, data curation, formal analysis, funding acquisition, investigation, methodology, and writing—review and editing.

K Lawlor: conceptualization, data curation, formal analysis, investigation, methodology, and writing—review and editing.
H Schlueter: conceptualization, formal analysis, investigation, methodology, and writing—review and editing.
Z Pieterse: data curation, formal analysis, investigation, and methodology.
Y Yu: conceptualization.
P Kaur: conceptualization, data curation, formal analysis, supervision, funding acquisition, validation, investigation, methodology, project administration, and writing—original draft, review, and editing.

### Conflict of Interest Statement

The authors declare that they have no conflict of interest.

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
