## [Reviewer comments · Life Science Alliance]

Pericytes promote skin regeneration by inducing epidermal cell polarity and planar cell divisions

Lizhe Zhuang, Kynan Lawlor, Holger Schluter, Zalitha Pieterse, Yu Yu and Pritinder Kaur.

DOI: 10.26508/lsa.201800009

Review timeline:

Submission Date:	8 December 2017
Editorial Decision:	31 January 2018
Revision Received:	14 June 2018
Additional Correspondence:	6 July 2018
Revision Received:	9 July 2018
Editorial Decision:	11 July 2018
Accepted:	17 July 2018

Report:

(Note: Letters and reports are not edited. The original formatting of letters and referee reports may not be reflected in this compilation.)

1st Editorial Decision

31 January 2018

Thank you for submitting your manuscript entitled "Pericytes promote symmetric cell divisions during human skin regeneration" to Life Science Alliance. The manuscript was assessed by expert reviewers, whose comments are appended to this letter. We invite you to submit a revision if you can address the reviewers' key concerns, as outlined here.

Importantly, the following concerns need to be addressed:

Ref#1:

- revise text accordingly

Ref#2:

- to support the claim of symmetric divisions, either do lineage tracing or at least use markers and static analyses
- unbiased quantifications are needed (specific comments point 1; see also ref#3's comments)
- specific comments point 3, 5 and 6 need to be addressed (see also ref#3 for comments on BMP2)
- specific comment 2 and 4 needs to be discussed

Ref#3:

- similar to Ref#2; please address quantification issues and provide further evidence for the role played by BMP2

Thank you for this interesting contribution to Life Science Alliance. We are looking forward to receiving your revised manuscript.

REFEREE REPORTS

Reviewer #1 (Comments to the Authors (Required)):

In their manuscript, Zhuang et al. show the importance of dermal microvascular pericytes in epidermal regeneration - including epidermal morphology, extracellular matrix deposition, cell proliferative activity, and symmetric cell division. Interestingly, they show that BMP2, a secreted morphogenetic factor expressed by dermal pericytes, could function as a paracrine regulator of symmetric keratinocyte cell division. Together, these descriptive observations are interesting for better understanding the regulatory mechanisms that instruct epidermal self-renewal. In general, the data they presented in the manuscript support their conclusion. There are several minor errors such as typos in the text which need to be corrected.

1. Last sentence in the first paragraph of the Discussion section is too long, and does not convey their message clearly
2. The first letter "w" in "wnt/BMP/FGF...", the last paragraph on the page 14, and throughout the page 15, should be replaced by capital "W".

Reviewer #2 (Comments to the Authors (Required)):

The authors investigate an interesting and rather poorly understood topic of how the skin microenvironment regulates its homeostasis. Specifically, the authors address the role of fibroblasts and pericytes in epidermal division and differentiation using human organotypic raft cultures as a model system. Using organotypic cultures (OC) the authors compare the ability of neonatal fibroblasts and pericytes in supporting the formation of a proper epidermis and find that pericytes in the absence of fibroblasts are the best source for this. They then provide data that indicate that pericytes control epidermal-dermal junction formation, spindle orientation (they use the term symmetric division but as they do not directly trace the cell fate of these cells nor use cell fate markers, the term is misplaced here) and epidermal maturation likely through paracrine signaling via BMP-2 using organotypic raft cultures as their model system.

As most people use cultured fibroblasts but not primary isolated cells in OCs, the novelty of this paper is the use of primary fibroblasts and pericytes in these cultures to assess their function and find that pericytes are able to better support the formation of histologically more normal looking OCs. As these OCs are used quite a bit not only to assess biology but also for therapy, this is an important finding.

The data on BMP2 are a little less convincing and many of the data are not properly quantified but based on one image shown.

Specific comments.

1. As many of the established OC systems are done with cultured fibroblasts, it would have been very worthwhile to compare pericytes alone to one of the established systems for a direct comparison, as the data suggest that pericytes might be the better cell type to get reproducible results. In this light, the authors compare the different combinations of OCs to human skin sections but only in the histology but not in their quantification. The argument that pericytes are the most comparable to human skin would be much stronger if the comparison would also be done in terms of quantification. This also in light of the fact that it has been shown that protocols to generate OCs with e.g. mouse fibroblasts and human keratinocytes (e.g. Arnette et al., *Methods in Enzymology*, 2015), show normal K10 expression levels in the absence of pericytes.

2. The authors identify pericytes as important components of the epidermal niche. Co-cultures of pericytes with previously used fibroblasts reveal changes in the epidermis such as disorganized basal layer and lower Δ N-P63 expression which are improved by OCs generated only with pericytes. This provokes the question whether fibroblast somehow inhibit pericyte function. Does coculture of pericytes and fibroblasts reduce secretion of key factors?

3. The data on BMP2 are not very strong. The authors do not show that the OCs look much better but only provide a close up of the basal layer. Is the BM also better? More importantly, looking at the supplementary data the increase in BMP2 RNA is less than 1.5 fold and based on what the authors show in A of the same supplementary figure this does not really seem to increase BMP2

protein expression or secretion. The authors should show that these cells indeed secrete substantially more BMP2 that is comparable to what pericytes secrete.

4. The observation that fibroblasts and pericytes might have differential localization in the skin, and that this localization is perhaps crucial to explain the difference in affecting epidermal tissue organization, renewal and differentiation is interesting. For physiological relevance it would be important to show that this difference in localization also occurs in vivo in human skin, also in light of recent findings that there are functionally and spatially different fibroblast skin populations.

5. The fact that fibroblast cultures show lower levels of K10 compared to pericytes cultures complicates the idea that fibroblasts support differentiation whereas pericytes promote self-renewal. To better understand this, it could be helpful to include other differentiation markers such as loricrin in the analysis.

6. Quantification and number of replicates need to be better done/described, see comments to individual figures.

Fig. 1

- They claim to observe thicker epidermis in F or F+P conditions, however, this has not been confirmed by any quantification. This is important as OCs normally exhibit highly variable thicknesses even under the same conditions

Fig. 2:

- Quantifications are missing

- Data obtained from 2 independent experiments. When analyzing OCs the minimum of 3 independent experiments are required and therefore 2 replicates are not enough to reach precise conclusions.

Fig. 3

- Proper spindle orientation is maintained in late metaphase (Poulson et al., JCB, 2010), therefore, it is important to analyze late stage mitotic cells. In this manuscript, it seems that cells in earlier mitotic stages have been analyzed. Here it is helpful to visualize DNA as well as the spindle midbody marker surviving, thus allowing better assessment of later mitotic stages of the cells.

- It is not clear how planar and perpendicular divisions were assessed. In panel (I) authors claim 10% of cells divide in a plane parallel to the surface in condition F, which is not clear from the radial bar charts presented.

- Overall, to avoid misinterpretation of spindle angles, at least 50 mitotic cells must be counted and the numbers of mitotic cells counted here 43, 23 and 23 per conditions are too low.

- Discrepancy between the text and figure legend. Page 9, authors claim the division angles were obtained from 4 independent experiments whereas in the figure legend (I) they mention 3.

Fig. 4

- The number of independent experiments and counted mitotic cells are not mentioned.

Reviewer #3 (Comments to the Authors (Required)):

This paper focusses on the role of pericytes in the epidermal stem cell niche using organotypic cultures of human epidermis. Understanding the cell and molecular components of the niche is an important problem, and culture offers a way in to resolve some aspects of the niche that are

technically infeasible to dissect in vivo.

The basis of this work is that pericytes may be reliably sorted by ITGA1 and CD45 expression. These are seeded onto transwell inserts and after a week, keratinocytes are added. It would be very helpful to cartoon the protocol to save readers having to look up the 2007 paper. Cultures with pericytes alone, fibroblasts alone and a mixture of both cell types are compared. It is important to know what happens if neither cell type is included, as organotypic cultures can be successfully generated on 'dead' dermis with no viable dermal cells.

Cultures are compared by immunostaining of histological sections. This can be problematic as usually organotypic epidermis varies across the culture dish. One way round this is top down imaging of the cultured sheet. If feasible, this sort of analysis, would support the conclusions more robustly. The presence of pericytes increases the proportion of proliferating basal cells as assessed by Ki67 staining. The argument that KRT10 expression is more 'ordered' is difficult to assess based on the data provided, none of the images shown closely resemble epidermis, a common feature of this type of culture. The basement membrane is argued to be different in the presence of pericytes, but the limited imaging and lack of quantifiable features in the images shown make it difficult to be sure how representative they are and whether any differences have functional significance.

Pericytes do tilt the angle of cell division parallel to the basement membrane, which is important as this is what occurs in vivo. It would be helpful to add a short-term lineage trace after a pulse of low dose EdU or BrdU to capture the fate of cells after division. Over expression of BMP2 in fibroblasts also increased the proportion of divisions parallel with the basement membrane. Knockdown or Crispr deletion of BMP2 in pericytes would provide confirmation of the role of BMP2 in the phenotype.

In discussing these results, it is important to stress that cells in divisions parallel to the basement membrane may have different fates.

Overall there are some interesting observations here but additional evidence is needed to draw firm conclusions.

1st Revision – authors' response

14 June 2018

We thank all 3 reviewers for their valuable comments and suggestions to improve our manuscript. We have made significant changes to the revised version of this manuscript in response to the critique provided and include significant new data requested by the referees to improve quantitative aspects, provide data from replicate experiments and provided additional data to support the role of BMP-2 in promoting keratinocyte divisions. Specifically we have added quantitation to Figure 1 & 4, added new panels to Fig 4, modified Fig 3 to include DAPI staining for mitotic figures, included a new Supplementary Figure 2 in response to concerns about the reproducibility of the data on keratinocyte polarisation and EM analysis in OCs from replicate experiments, and included a modified Supplementary Figure 3 (formerly Suppl Figure 2) to include BMP-2 protein data, in addition to data for a new marker K15 which is indicative of homeostasis in basal keratinocytes. We have revised the manuscript text (including figure legends) throughout to reflect these changes all marked in red, and included comments as suggested by reviewers or that otherwise provide further explanations for our rationale. Finally, the manuscript layout has been re-formatted according to the Instructions for Authors for Life Science Alliance. Please find below, a detailed point-by-point response to each of the Reviewers' critique.

Reviewer #1:

There are several minor errors such as typos in the text which need to be corrected.

1. Last sentence in the first paragraph of the Discussion section is too long, and does not convey their message clearly.

Edited for clarity

2. The first letter "w" in "wnt/BMP/FGF...", the last paragraph on the page 14, and throughout the page 15, should be replaced by capital "W".

Changed as requested.

Reviewer #2:

General Comments:

They then provide data that indicate that pericytes control epidermal-dermal junction formation, spindle orientation (they use the term symmetric division but as they do not directly trace the cell fate of these cells nor use cell fate markers, the term is misplaced here) and epidermal maturation likely through paracrine signaling via BMP-2 using organotypic raft cultures as their model system.

We believe that the use of the term symmetric division is well-justified given that we have followed up the composition of the basal layer cells with several markers (p63, Ki67, K10 & now K15, a marker known to mark homeostatic basal epidermal cells) which demonstrate that cultures that contain a higher frequency of cell divisions parallel to the epidermal:dermal junction (pericyte co-cultures) are maintained in a more proliferative state whereas those with fewer parallel divisions do not (fibroblasts or fibroblasts+pericytes). While we agree that the ideal scenario would be to trace the fate of cells dividing at different angles – this is technically beyond current state-of-the-art technologies, not to mention a big ask in 3D organotypic culture given that the collagen gels lack opacity and the need to introduce stable fluorescent labels to track individual cells – not a realistic option when working with primary freshly isolated keratinocytes. We have added a paragraph to the Discussion to reflect these considerations and indeed provide supporting evidence from the literature from notable skin labs (Lechler, Fuchs, Greco) that strongly indicate that cells do not return to the basal layer once they migrate suprabasally.

The data on BMP2 are a little less convincing and many of the data are not properly quantified but based on one image shown.

On re-reviewing the manuscript, we realise that we did not specifically mention how many replicate experiments the BMP-2 data was derived from. This was an oversight since three replicate experiments were performed as stated in the Quantification and statistics section of the Materials and Methods: “Quantitative results from immunostaining and OC experiments were replicated in at least 3 independent experiments, and representative samples displayed when reproducible results were obtained”. We have revised the manuscript to indicate that the data shown in Fig 4 are derived from 3 independent experiments in both the Results section and Fig 4 legend. We also include quantitation in the revised Figure 4 showing data for thickness of the reformed epithelium (panel L), number of Ki67+ cells (M) and number of ΔNp63+ cells in GFP-control and BMP-2 transduced fibroblasts from 3 replicate experiments. We apologise for not including this essential data in the original manuscript – clearly a major oversight on our part.

Reviewer #2, Specific comments:

1.a. As many of the established OC systems are done with cultured fibroblasts, it would have been very worthwhile to compare pericytes alone to one of the established systems for a direct comparison, as the data suggest that pericytes might be the better cell type to get reproducible results.

The reviewer requests a direct comparison with established systems suggesting the use of cultured fibroblasts – this comment is puzzling given that we have indeed used cultured fibroblasts (against pericytes) for comparison in the organotypic cultures.

1.b. In this light, the authors compare the different combinations of OCs to human skin sections but only in the histology but not in their quantification. The argument that pericytes are the most comparable to human skin would be much stronger if the comparison would also be done in terms of quantification.

We have deleted reference to the pericyte regenerated epidermis being “most comparable to human skin” on Page 7 of the Results and modified our statement later on the same page to read “with the closest similarity to native skin”. Not surprisingly, the OC model system only approximates native human skin given that many elements of native skin are not included in these cultures (nerves,

immune cells, melanocytes etc). We would stress that our observations are qualitative i.e. spatial and temporal gene expression, and not quantitative.

1.c. This also in light of the fact that it has been shown that protocols to generate OCs with e.g. mouse fibroblasts and human keratinocytes (e.g. Arnette et al., Methods in Enzymology, 2015), show normal K10 expression levels in the absence of pericytes.

We do not claim that pericytes are essential for this. Indeed, our data shows K10 expression in organotypics containing either fibroblasts or pericytes (Fig 1D). Rather As stated on Page 7 of the results, we claim “the most ordered spatial expression of the differentiation-specific keratin K10”.

2. The authors identify pericytes as important components of the epidermal niche. Co-cultures of pericytes with previously used fibroblasts reveal changes in the epidermis such as disorganized basal layer and lower Δ N-P63 expression which are improved by OCs generated only with pericytes. This provokes the question whether fibroblast somehow inhibit pericyte function. Does coculture of pericytes and fibroblasts reduce secretion of key factors?

The reviewer is very observant noting potential effects of co-culturing keratinocytes with fibroblasts and pericytes. This does appear to be the case and the next step in this work would be to investigate alterations in secreted factors – however this is beyond the scope of the current study. We have added the need to do this in the Discussion.

3.a. The data on BMP2 are not very strong. The authors do not show that the OCs look much better but only provide a close up of the basal layer. Is the BM also better?

We apologise for not including lower magnification images of the organotypic cultures with BMP-2 transduced HFFs – an unacceptable omission on our part. Figure 4, Panel A has now been revised to include both low power and high power magnifications revealing that the entire epithelial layer is thickened in addition to the changes we reported in the basal layer. Moreover, we now include quantitative data on epidermal thickness (Fig 4L), number of Ki67+ cells (Fig 4 M) and number of basal Δ Np63+ cells (Fig 4N previously described as data not shown), which substantiate our claims. With respect to the BM, as shown in Fig 4D and discussed in the original manuscript on page 10, the BM is improved but not to the extent seen with pericytes. “Immunostaining revealed increased LAMA5 deposition at the dermo:epidermal junction...(Figure 4D...) in OCs with F_BMP2 fibroblasts compared to F_GFP controls – although, this was not restored to the levels observed with pericyte inclusion in the OCs (Figure 1D and 2A),” We did not look at EM level since similar LAMA5 staining does not correlate with BM assembly (as shown in Fig 2B).

3.b. More importantly, looking at the supplementary data the increase in BMP2 RNA is less than 1.5 fold and based on what the authors show in A of the same supplementary figure this does not really seem to increase BMP2 protein expression or secretion. The authors should show that these cells indeed secrete substantially more BMP2 that is comparable to what pericytes secrete.

While it is true that the amount of BMP-2 mRNA is only increased 1.5 fold in the HFF-BMP2 compared to the HFF-GFP controls, this level approximates BMP-2 mRNA levels in pericytes (Paquet-Fifield et al., 2009), and is likely to be physiologically relevant. As with most cytokines high level expression is not required for function. The reviewer has mistakenly assumed that Supplementary Figure 2A shows BMP-2 expression. In fact, this panel shows equivalent GFP expression in both control (F_GFP) and BMP-2 transduced cells (F_BMP2) to illustrate that retroviral transduction was successful in both types of fibroblasts. We had anticipated that given the low levels of BMP-2 mRNA in pericytes, it would be difficult to demonstrate BMP-2 secretion at protein level. However, since receiving the Reviewers’ comments, we have more thoroughly investigated levels of BMP-2 protein expression in pericytes as well as control and BMP-2 transduced fibroblasts. We conclude that the levels of BMP-2 expression are very low in all cells examined. Notably, even in pericytes, we needed to block secretion of BMP-2 by treating cells with Brefeldin A (which results in accumulation of secreted proteins in the Golgi), to clearly visualise its presence by in situ immunofluorescence (IMMF). Quantitation of the IMMF signal/cell showed that:

- pericytes express BMP-2 protein although every single cell is not positive.*
- lower levels of BMP-2 protein are detectable in F_BMP2 compared to pericytes*
- background to negative levels of BMP-2 were detected in control F_GFP fibroblasts.*

These data are included in revised Supplementary Figure 3. We conclude that the level of BMP-2 is critical for biological effect and that the BMP-2 transduced fibroblasts make sufficient amounts of this morphogen to have an impact on keratinocyte proliferation and spindle orientation, whereas the levels expressed by control fibroblasts if they can be considered to be positive at all, are insufficient. This is consistent with the reported ability of morphogens such as BMP-2 to have their impact in a dose-dependent manner creating cytokine gradients in whole organisms. Our previous microarray analysis showed preferential expression of BMP-2 in pericytes compared to fibroblasts (Paquet-Fifield et al., 2009)- however that analysis was performed in freshly isolated, uncultured pericytes and fibroblasts. We infer that BMP-2 expression may be induced in cultured fibroblasts, but clearly at a level that is ineffective biologically as evidenced by the lack of impact on keratinocytes in organotypic cultures.

4. The observation that fibroblasts and pericytes might have differential localization in the skin, and that this localization is perhaps crucial to explain the difference in affecting epidermal tissue organization, renewal and differentiation is interesting. For physiological relevance it would be important to show that this difference in localization also occurs in vivo in human skin, also in light of recent findings that there are functionally and spatially different fibroblast skin populations.

Efforts to demonstrate differential localisation of pericytes versus fibroblasts are ongoing in our laboratory and while we have established 3D imaging techniques to visualise these cells in whole mounts of skin, we are still a long way from generating quantitative data, largely complicated by difficulty with reagents that despite working well in tissue sections, fail to do so in whole mounts. Certainly, this is an important question that remains a goal of our current work, but beyond the scope of the current manuscript.

5. The fact that fibroblast cultures show lower levels of K10 compared to pericytes cultures complicates the idea that fibroblasts support differentiation whereas pericytes promote self-renewal. To better understand this, it could be helpful to include other differentiation markers such as loricrin in the analysis.

We disagree with this simplistic interpretation – a better conclusion is that pericytes promote self-renewal and differentiation, whereas fibroblasts do not promote self-renewal, but as reported previously, support differentiation (El Ghalbzouri & Ponc, Cell Tissue Res (2002) 310:189–199). Thus, analysis of additional differentiation markers is unlikely to add value to our data which deliberately focus on changes to the basal layer where the most profound changes are elicited with pericyte co-culture.

6.a. Fig. 1. They claim to observe thicker epidermis in F or F+P conditions, however, this has not been confirmed by any quantification. This is important as OCs normally exhibit highly variable thicknesses even under the same conditions.

We had this data but did not include it in the original manuscript thinking that epidermal thickness was not as valuable as quantitation of specific proliferative markers. We now include this quantification in Figure 1H for OCs which show reproducible results from pooling the data from 3 independent experiments.

6.b. Fig. 2 - Quantifications are missing. Data obtained from 2 independent experiments. When analyzing OCs the minimum of 3 independent experiments are required and therefore 2 replicates are not enough to reach precise conclusions.

As stated in the Figure legend to Fig 2 in the original submission, the results for LAMA5 staining are representative of 3 independent experiments, not 2; the ultrastructural TEM analysis for basement membrane and hemidesmosome assembly was indeed from 2 independent experiments – given that the data were highly reproducible between replicates, we believe this is sufficient to draw the conclusion that BM assembly occurs only in pericyte co-cultures since this data is complementary to the 3 separate independent experiments shown in Fig 2, panel A. Nevertheless, we now include more representative images from the two replicate experiments in a new Supplementary Fig 2, panel C to illustrate the reproducibility of the TEM data. Moreover, we also include additional images of histological sections of OCs from each replicate experiment (Suppl Fig 2, panel B) to illustrate that we obtained reproducible results across all three replicate experiments

with respect to polarisation of the basal layer when co-cultured with pericytes compared to fibroblasts alone or fibroblasts+pericytes.

6.c. Fig. 3 - Proper spindle orientation is maintained in late metaphase (Poulson et al., JCB, 2010), therefore, it is important to analyze late stage mitotic cells. In this manuscript, it seems that cells in earlier mitotic stages have been analyzed. Here it is helpful to visualize DNA as well as the spindle midbody marker surviving, thus allowing better assessment of later mitotic stages of the cells. Overall, to avoid misinterpretation of spindle angles, at least 50 mitotic cells must be counted and the numbers of mitotic cells counted here 43, 23 and 23 per conditions are too low.

We did not select any particular phase of mitoses for the spindle orientation analyses but selected PH3 positive cells to identify mitotic cells – it is virtually impossible to find enough mitotic cells in a specific stage, let alone the ideal number of 50 mitoses per condition analysed. Mitotic cells are rare in 3D organotypic cultures, particularly at the time point we analysed. We agree that sections should be stained for PH3/Y-tubulin and DAPI – we neglected to mention that we had done exactly that. We have modified Fig 3 to include the DAPI stain in a new panel (G). Our painstaking analysis very clearly show differences in frequency of cell division angles with a pericyte environment despite the low numbers available to us for analysis in these type of cultures. We would argue that the quantitative molecular evidence i.e. increase in % p63/Ki67 cells and the increase in LAMA5 and basal K15 expression in pericyte co-cultured OCs, clearly supports the notion that more cells are retained in the basal layer under these conditions.

6.d. It is not clear how planar and perpendicular divisions were assessed. In panel (I) authors claim 10% of cells divide in a plane parallel to the surface in condition F, which is not clear from the radial bar charts presented.

We state that “Analysis of the percentage of mitoses against the angle of division orientation (0° to 90°) revealed that co-culture with pericytes resulted in ~50% of mitoses dividing in a plane parallel to the dermo-epidermal junction (<10° angle)compared to ~10% in fibroblast co-cultures. Below we show how we arrived at ~10%. On the left is the panel shown in the manuscript, and on the right, the same panel marked up to illustrate how it is read. The blue wedge indicates all mitoses scored as occurring at an angle of 0-10o in the radial display in F co-cultures. The bar value is extrapolated to the y-axis in an arc (red) and the corresponding value read. This is a standard approach used by many authors to depict mitotic spindle orientation – but takes a bit of getting used to.

6.e. Discrepancy between the text and figure legend. Page 9, authors claim the division angles were obtained from 4 independent experiments whereas in the figure legend (I) they mention 3.

We thank the reviewer for drawing our attention to this error. The data were in fact from 4 independent experiments – Figure 1 legend has been revised to reflect this.

6.f. Fig. 4, The number of independent experiments and counted mitotic cells are not mentioned.

Our apologies for this unintentional omission. The number of experiments (n=3) and the number of mitotic events counted (n= 28, for F_GFP and n=27 for F_BMP-2) are now stated in the Figure 4 legend.

Reviewer #3:

1. The basis of this work is that pericytes may be reliably sorted by ITGA1 and CD45 expression. These are seeded onto transwell inserts and after a week, keratinocytes are added. It would be very helpful to cartoon the protocol to save readers having to look up the 2007 paper.

We now include a new Supplementary Figure 2 in which we include a cartoon in panel A as requested.

2. Cultures with pericytes alone, fibroblasts alone and a mixture of both cell types are compared. It is important to know what happens if neither cell type is included, as organotypic cultures can be successfully generated on 'dead' dermis with no viable dermal cells.

The proposed experiment has actually been published by others (El Ghalbzouri & Ponec, Cell Tissue Res, 2002, 310:189–199), resulting in a poorly formed epidermis with 2-3 viable cell layers with flattened nuclei in an indistinct “basal layer”, a very thin stratum corneum and lacking a distinct stratum granulosum on a dead dermis. Indeed this paper very nicely illustrated the dependency of keratinocytes on fibroblasts and their growth factors for proper epidermalisation and differentiation. It seemed logical to us therefore, to compare the effect of different types of mesenchymal cells rather than no mesenchymal support at all given that this always results in poor skin regeneration.

3. Cultures are compared by immunostaining of histological sections. This can be problematic as usually organotypic epidermis varies across the culture dish. One way round this is top down imaging of the cultured sheet. If feasible, this sort of analysis, would support the conclusions more robustly.

It is not clear why this approach is being suggested – we are not aware of this kind of approach being used or indeed proven to give better results. We are well aware of the variability across the culture dish – and that is why we have performed many replicates in each experiment as well as several independent repeat experiments. Quantitative analysis across 3 independent experiments have yielded data that are statistically robust and stand up to close scrutiny across experiments. The protocol we use was developed by Organogenesis Inc, Boston, MA and was extensively optimised by them for quality control and reproducibility given its intended use as a therapeutic biologic in the clinic. This method is more complex than those utilised by most labs using specialised media at different stages of the culture. In our hands, using the Organogenesis protocol combined with the bovine collagen supplied by them has always given reproducible, reliable data across many of our publications (Li et al., J Clin Invest 2004; Paquet-Fifield et al., J Clin Invest 2009).

4. The presence of pericytes increases the proportion of proliferating basal cells as assessed by Ki67 staining. The argument that KRT10 expression is more 'ordered' is difficult to assess based on the data provided, none of the images shown closely resemble epidermis, a common feature of this type of culture.

We respectfully disagree with this comment – the data we show very clearly illustrates uniform expression of K10 in the suprabasal layers in pericyte co-cultures – this is not the case for fibroblast or fibroblast+pericyte co-cultures. Our contention is that pericytes induce correct spatial expression, which is what we mean by ordered K10 expression. We have revised the wording to read “less uniform” rather than “disordered” on page 7.

5. The basement membrane (BM) is argued to be different in the presence of pericytes, but the limited imaging and lack of quantifiable features in the images shown make it difficult to be sure how representative they are and whether any differences have functional significance.

We agree that the figure we included prints poorly and may not have been clear enough to support our claim. We now include additional images from other independent experiments in the new Supplementary Figure 2 (panel C), supporting the veracity of our claim. The changes we observed (presence of BM and hemi-desmosomes) are qualitative i.e. either absent (fibroblast co-cultures) or present (pericyte co-cultures) – and therefore do not necessitate quantitation.

6. Pericytes do tilt the angle of cell division parallel to the basement membrane, which is important as this is what occurs in vivo. It would be helpful to add a short-term lineage trace after a pulse of low dose EdU or BrdU to capture the fate of cells after division.

This is a technically challenging task and can really only be addressed by live imaging of fluorescently tagged cells – something that remains beyond the grasp of any laboratory at present. See also response to Reviewer 2, point 6c above. Moreover,

7. Over expression of BMP2 in fibroblasts also increased the proportion of divisions parallel with the basement membrane. Knockdown or Crispr deletion of BMP2 in pericytes would provide confirmation of the role of BMP2 in the phenotype.

We agree that this would further validate our finding, but in view of a similar role for BMP-2 in other model organisms (mice and flies) presented in the Discussion Page 15 (Zhu et al., 2014; Kawase et al. 2004) , we feel confident that the evidence we provide adequately supports our argument.

8. In discussing these results, it is important to stress that cells in divisions parallel to the basement membrane may have different fates.

We have revised the discussion to include this statement.

2nd Editorial Decision

6 July 2018

Thank you for submitting your revised manuscript entitled "Pericytes promote symmetric cell divisions during human skin regeneration." to Life Science Alliance. The manuscript was assessed by two of the original reviewers again, whose comments are appended to this letter.

As you will see, reviewer #2 thinks that in absence of lineage tracing, the term 'symmetric cell division' should not be used as this can be interpreted to the establishment of daughter cells with equal fate, which may or may not be the case in your OCs. We agree with this view. Reviewer #3 is disappointed by the revision and stated to us that s/he does not support publication of your revised manuscript here. Importantly, this reviewer points out that neither lineage tracing nor BMP2 knock-down/knock-out has been performed, experiments we asked you to include to allow publication of your work in Life Science Alliance. Your argument that BMP2 has been previously linked to stem cell niche regulation in *Drosophila* and to proliferation in the dermis of mice is in our view not sufficient. Therefore, I am afraid, we concluded that we cannot offer publication at this stage.

We realize that reviewer #2 (and reviewer #1 of the previous round of review) support publication of a further revised manuscript. Though we usually only allow a single round of review, we can invite you to submit a further revised version should you be able to include BMP2 knock-down analyses and re-write the manuscript to refer to planar cell divisions instead of symmetric divisions. If you decide to do so, please also consider re-ordering your figure panels for a better chronological order.

Thank you for this interesting contribution to Life Science Alliance. We are looking forward to receiving your revised manuscript.

REFeree REPORTS

Reviewer #2 (Comments to the Authors (Required)):

This paper is now sufficiently improved and provides very relevant and important insight into the role of pericytes in epidermal/skin homeostasis.

I do still disagree that they call the changes in spindle orientation increased symmetric divisions. To claim symmetric division or asymmetric divisions is showing that the division itself is directly resulting in either similarly fated daughters or differentially fated daughters. This is not done in this paper, as indeed to show direct division-mediated fate changes is hard to show and I do not expect them to do that. However, what they examine is spindle angles and this is linked/accompanied by increased expression of basal markers, indeed suggestion perhaps even indicating that these may be

symmetric divisions. It is still possible that pericytes promote basal fate through different mechanisms that are independent of increased planar divisions, including e.g. secretion of growth factors. Moreover, as also pointed out by the other reviewer there is enough evidence in the literature that symmetric divisions or better said planar spindles/planar divisions can give rise to an asymmetric fate. I do consider it important to distinguish between the two terms. Importantly for me this distinction does not take anything away from the interest of their findings nor the important role of pericytes in skin biology that is revealed here.

Reviewer #3 (Comments to the Authors (Required)):

The authors have made some improvements to the manuscript in response to the reviewers. The lack of a BMP knockdown experiment remains a weakness in the manuscript and it is disappointing that short term EdU lineage tracing was not performed, as this does not require live imaging equipment.

Additional Correspondence - Editor

6 July 2018

Many thanks for the phone call following our decision to ask you to further revise your work.

We appreciate your explanation that for BMP2 depletion analysis CRISPR-mediated knock-out would be required given the timeframe of organotypic culture (OC) experiments. We also understand that you hadn't done a knock-out experiment for BMP2 nor EdU lineage tracing in organotypic skin cultures because you cannot receive the same OC starting material anymore and are currently performing laborious and time-consuming optimization procedures to set-up the system again.

I have re-discussed your work explaining the circumstances with an academic editor. Because reviewer #1 and #2 were positive about your work even in absence of BMP2 knock-out experiments, we decided to overrule our request. We would, however, expect that you clearly state in the manuscript text that more definitive proof for the role of BMP2 is needed, and that you tone-down your conclusions accordingly. The other revision requests (symmetric division; chronological order of figure descriptions) should be performed as already outlined in the original decision letter. Please note that we have the 'Follow Up' format at Life Science Alliance that may allow adding BMP2 knock-out data at a later time.

2nd Revision – authors' response

9 July 2018

1. "...clearly state in the manuscript text that more definitive proof for the role of BMP2 is needed, and that you tone-down your conclusions accordingly".

We have revised the manuscript to do just this in the Abstract, Summary Blurb and in the Discussion where we included the sentence "Further, unequivocal evidence in support of a role for BMP2 in influencing spindle pole or cell division orientation in keratinocytes, such as BMP2 knockdown in pericytes is essential".

2. "reviewer #2 thinks that in absence of lineage tracing, the term 'symmetric cell division' should not be used as this can be interpreted to the establishment of daughter cells with equal fate, which may or may not be the case in your OCs".

We have replaced the term "symmetric" with "planar" throughout the manuscript when referring to the results obtained. In addition, we have taken the liberty of revising the title of the manuscript from "Pericytes promote symmetric cell divisions during human skin regeneration." To "Pericytes promote human skin regeneration by inducing epidermal cell polarity and planar cell divisions", which we hope is more acceptable.

3. "...please also consider re-ordering your figure panels for a better chronological order."

We have re-arranged panels in Figures 1 & 4 to allow us to cite each panel sequentially. Two sections of the Results section have been re-written to allow appropriate sequential reference to panels in Suppl Fig 4. All text affected by the panel renaming has been altered throughout the manuscript (Results & Discussion), and the Figure legends revised accordingly.

3rd Editorial Decision

11 July 2018

Thank you for submitting your revised manuscript entitled "Pericytes promote skin regeneration by inducing epidermal cell polarity and planar cell divisions". I appreciate the introduced changes and am happy to accept your manuscript in principle for publication in Life Science Alliance.
